# Immune Response after mRNA COVID-19 Vaccination in Lung Transplant Recipients: A 6-Month Follow-Up

**DOI:** 10.3390/vaccines10071130

**Published:** 2022-07-15

**Authors:** Selma Tobudic, Alberto Benazzo, Maximilian Koblischke, Lisa Schneider, Stephan Blüml, Florian Winkler, Hannah Schmidt, Stefan Vorlen, Helmuth Haslacher, Thomas Perkmann, Heinz Burgmann, Peter Jaksch, Judith H. Aberle, Stefan Winkler

**Affiliations:** 1Division of Infectious Diseases, Department of Internal Medicine I, Medical University of Vienna, 1090 Vienna, Austria; lisa.schneider@meduniwien.ac.at (L.S.); n1613753@students.meduniwien.ac.at (F.W.); n11707205@students.meduniwien.ac.at (H.S.); n11718466@students.meduniwien.ac.at (S.V.); heinz.burgmann@meduniwien.ac.at (H.B.); stefan.winkler@meduniwien.ac.at (S.W.); 2Clinical Division of Thoracic Surgery, Department of Surgery, Medical University of Vienna, 1090 Vienna, Austria; alberto.benazzo@meduniwien.ac.at (A.B.); peter.jaksch@meduniwien.ac.at (P.J.); 3Center for Virology, Medical University of Vienna, 1090 Vienna, Austria; maximilian.koblischke@meduniwien.ac.at (M.K.); judith.aberle@meduniwien.ac.at (J.H.A.); 4Division of Rheumatology, Department of Internal Medicine III, Medical University of Vienna, 1090 Vienna, Austria; stephan.blueml@meduniwien.ac.at; 5Department of Laboratory Medicine, Medical University of Vienna, 1090 Vienna, Austria; helmuth.haslacher@meduniwien.ac.at (H.H.); thomas.perkmann@meduniwien.ac.at (T.P.)

**Keywords:** lung transplant recipients, T cell response, humoral response, immunosuppressive drugs, COVID-19

## Abstract

Background and objective: This prospective cohort study analyzed the immune response to COVID-19 mRNA vaccines in lung transplant recipients (LuTRs) compared to healthy controls (HCs) at a 6-month follow-up. Methods: After the first two doses of either BNT162b2 or mRNA-1273, SARS-CoV-2 antibodies were measured in LuTRs (*n* = 57) and sex- and age-matched HCs (*n* = 57). Antibody kinetics during a 6-month follow-up and the effect of a third vaccine dose were evaluated. Humoral responses were assessed using the Elecsys^®^ Anti-SARS-CoV-2 S immunoassay. In 16 LuTRs, SARS-CoV-2-specific T cell responses were quantified using IFN-γ ELISpot assays. Results: Seroconversion rates were 94% and 100% after the first and second vaccine dose, respectively, in HCs, while only 19% and 56% of LuTRs developed antibodies. Furthermore, 22 of 24 LuTRs who received the third vaccine dose showed seroconversion (five of seven primary non-responders and 17 of 17 primary responders). A T cell response against SARS-CoV-2-spike S1 and/or S2 was detected in 100% (16/16) of HCs and 50% (8/16) of LuTRs. Conclusions: The data suggest that LuTRs have reduced humoral and cellular immune responses after two doses of COVID-19 mRNA vaccination when compared to HCs. A third dose may be of substantial benefit.

## 1. Introduction

Lung organ recipients (LuTRs) are at increased risk of severe COVID-19 infection, caused by a high instances of comorbid conditions and their immunocompromised status [1,2,3,4].

COVID-19 vaccines have been developed to combat the ongoing pandemic by preventing primary infection and severe disease courses. In healthy individuals, COVID-19 mRNA vaccines are highly immunogenic [5,6]. In contrast, immunocompromised individuals have reduced immune responses after primary vaccination, with a significant number of patients not seroconverting, leaving them more susceptible to SARS-CoV-2 infections and subsequent severe disease courses [7,8,9,10,11]. In solid organ transplant recipients (SOTRs), low antibody responses were associated with older age, use of anti-metabolite maintenance immunosuppressive therapies, and shorter time since transplantation [9,10,12].

It has become evident that protective immunity against SARS-CoV-2 is not of long duration, as evidenced by decreasing titers and increased breakthrough infection rates over time after initial vaccination, and a third vaccine dose was included in a standard SARS-CoV-2 vaccination program [13]. Recently published data in SOTRs have shown that a third dose of mRNA vaccine improves immunogenicity, as measured by SARS-CoV-2 spike antibody titers [14]. In contrast, lower vaccine-induced neutralization against SARS-CoV-2 variants of concern in SOTRs compared to healthy individuals was recently reported [15].

In this study, we analyzed immune responses after two doses of COVID-19 mRNA vaccines in LuTRs, including SARS-CoV-2 antibody titer kinetics, after 6 months, and immune responsiveness after the third vaccine dose.

## 2. Patients and Sample Collection

The study is part of a prospective cohort study performed at the Medical University of Vienna, Vienna, Austria, “Characterization of the responsiveness after SARS-CoV-2 mRNA vaccination in patients with immunodeficiency or immunosuppressive therapy”; Eudra CT Nr.2021-000291-11.

In this study, we prospectively enrolled 57 LuTRs from the Clinical Division of Thoracic Surgery outpatient ward. All patients were primarily vaccinated twice with a COVID-19 mRNA vaccine (BNT162b2, BioNTech/Pfizer or mRNA-1273, Moderna), adhering to the recommended time intervals between the two doses. At the 6-month follow-up, 24 LuTRs also received a third dose of the COVID-19 vaccine. As a control group, we selected 57 individuals (age- and sex-matched) from the original cohort of healthy controls (HCs). Antibodies against the SARS-CoV-2 receptor-binding domain and the nucleocapsid protein were determined before vaccination, 2–3 weeks after first immunization (median 17 days), 3–10 weeks after the second immunization (median 27 days), and 6 months after the second immunization (median 158 days). Serum samples for antibody tests were stored at the Biobank of the Medical University of Vienna (MedUni Wien Biobank), a centralized facility for preparing and storing biomaterials with certified quality management (ISO 9001:2015) [16]. Peripheral blood mononuclear cells (PBMCs) obtained 2–4 weeks after the second vaccination were isolated using density gradient centrifugation, and stored in liquid nitrogen until further use. Ethical approval for this study was granted by the ethics committee of the Medical University of Vienna, Austria (1291/2021). 

## 3. Anti-SARS-CoV-2 Testing

We used the Elecsys^®^ Anti-SARS-CoV-2 S immunoassay to determine the antibodies to the receptor-binding domain (RBD) of the viral spike (S) protein [17]. The quantitation range was between 0.4 and 2500.0 BAU/mL. Values > 0.8 BAU/mL were considered positive. Results below the lower level of quantification were defined as 0.2 BAU/mL to allow for calculations. Previous SARS-CoV-2 infection was declared by anamnesis, measuring nucleocapsid-specific antibodies with the qualitative Elecsys^®^ Anti-SARS-CoV-2 assay [18]. Antibody tests were performed on a cobas^®^ e801 analyzer (Roche Diagnostics, Rotkreuz, Switzerland) at the Department of Laboratory Medicine, Medical University of Vienna (certified acc. to ISO 9001:2015 and accredited acc. to ISO 15189:2012).

## 4. Peptides

For T cell stimulation, we used PepMix™SARS-CoV-2 peptide pools, purchased from JPT (Berlin, Germany). The pools cover the entire sequences of the SARS-CoV-2 spike protein and comprise 15-mer peptides with 11 amino acid (aa) overlaps. The spike peptides are split into two subpools, S1 (aa 1–643) and S2 (aa 633–1273). Peptides were dissolved in dimethyl sulfoxide and diluted in AIM-V medium for use in ELISpot assays [8].

## 5. IFN-γ ELISpot Assay

PBMCs were thawed and used for ex vivo ELISpot assays. A total of 1–2 × 10^5^ cells per well were incubated with SARS-CoV-2 peptides (2 μg/mL; duplicates), AIM-V medium (negative control; 3–4 wells), or PHA (L4144, Sigma, Kawasaki, Japan; 0.5 μg/mL; positive control), in 96-well plates coated with 1.5 μg anti-IFN-γ (1-D1K, Mabtech, Stockholm, Sweden) for 24 h. After washing, spots were developed with 0.1 μg biotin-conjugated anti-IFN-γ (7-B6-1, Mabtech), streptavidin-coupled alkaline phosphatase (Mabtech, 1:1000), and 5-bromo-4-chloro-3-indolyl phosphate/nitro blue tetrazolium (Sigma). Spots were counted using a Bio-Sys Bioreader 5000 Pro-S/BR177 (Miami, FL, USA) and Bioreader software generation 10. T cell responses were considered positive when mean spot counts were at least threefold higher than the mean spot counts of three unstimulated wells [8].

## 6. Statistical Analysis

Propensity scores were used to match both groups (control group, LuTR) by age and sex. This was implemented using the R package “MatchIt”. For the visit after the first vaccine dose, five values (9%) were missing for the humoral response variables in the LuTR group and eight values (14%) were missing in the control group. After the second vaccine dose, there were no missing values in the LuTR group and two (4%) in the control group. In the LuTR group, 19 values (33%) were missing 6 months after the third dose; in the control group, 31 values (54%) were missing. After the third dose, the humoral response variables contained 33 (58%) missing values in the LuTR group and four (7%) missing values in the control group. Missing values were omitted for analysis for each variable individually. According to the distribution, continuous variables are presented as the median value with interquartile range. Categorical variables of unpaired groups were compared using Fisher’s exact test. Numerical variables of unpaired groups were compared by the Wilcoxon rank-sum test. Differences in paired groups were compared using the chi square test or McNemar’s test for categorical variables, and the Wilcoxon matched pair rank-sum test for continuous variables. Bonferroni correction for multiple testing was applied when indicated. To assess factors that influence seroconversion, univariate logistic regression was applied. Continuous variables (age, time since transplant), and categorical variables (sex, diagnosis, therapy) were selected based on their expected relevance. Additionally, logistic regression was repeated with a multivariate model to adjust for age and sex. Statistical analysis was performed using R 4.1.1 (R Core Team (2021). Vienna, Austria). The R packages “ggplot2”, “ggpubr”, and “viridis” were used for graphical representation of the data.

## 7. Results

### 7.1. Patient Characteristics

A total of 114 study participants, 57 LuTRs (age, median; IQR 55 (46.5–58.25), 61% female) and 57 HCs (age, median; IQR 55 (44.0–64.0), 60% female) received two vaccinations with either BNT162b2 (Pfizer/BioNTech), *n* = 112 (98%), or mRNA-1273 (Moderna), *n* = 2 (2%). The median time between lung transplantation and COVID-19 vaccination was 7 (IQR 4–11) years. The third vaccination was given to 24 LuTRs at a median of 175 days (IQR 152–194) after the second dose. All LuTRs received immunosuppressive therapies in combinations of two (12%), three (75%), or four (12%) immunosuppressive drugs. Immunosuppressive medication included mTOR inhibitors (everolimus), *n* = 11 (19); calcineurin inhibitors, *n* = 56 (98%); tacrolimus, *n* = 55 (97%); ciclosporine, *n* = 1 (2%), mycophenolate mofetil/mycophenolic acid (MMF/MPA), *n* = 47 (58.2%); and glucocorticoids (prednisone 5 mg), *n* = 56 (98%) (Table 1).

### 7.2. Humoral Response after COVID-19 mRNA Vaccine

Humoral responses were analyzed in 114 study participants after the first two doses of COVID-19 mRNA vaccine. None of the LuTRs, and three HCs, had a clinical history of COVID-19 before the first vaccination. Participants with a clinical history of COVID-19 before the first vaccination were excluded from statistical analysis. Seroconversion rates of SARS-CoV-2 antibodies following COVID-19 mRNA vaccination were significantly lower in LuTRs than in HCs: 19% vs. 94% (*p* < 0.001) after the first immunization, and 56% vs. 100% (*p* < 0.001) after the second immunization. A significant increase in seroconversion rate was observed after the second compared to the first vaccine dose in LuTRs (*p* < 0.001). (Figure 1A). Median SARS-CoV-2 antibody levels were significantly lower after the first and second immunization in LuTRs compared to HCs (<0.4 (IQR <0.4–<0.4) vs. 19.6 (IQR 6.1–75.9), *p* < 0.001 and 3.96 (IQR < 0.2–133) vs. 1537.0 (820.5–2146), *p* < 0.001) (Figure 1B). Antibody levels increased significantly in LuTRs (*p* < 0.001) and HCs (*p* < 0.001) after the second vaccine dose (Figure 2A). At 5 months (IQR 4–6 months) after the second dose, SARS-CoV-2 S antibody levels were analyzed in 38 LuTRs and 27 HCs. Antibody titers increased in 14 LuTRs (median 122.7, IQR 39.5–323.3) and four HCs (median 244.5, IQR 136–570.8), and decreased in 24 LuTRs (median 162.1, IQR 8.9–564.8) and 21 HCs (median 881, IQR 632–1387). After the third dose of the COVID-19 vaccine, seroconversion was achieved in the LuTRs group in five of seven primary non-responders (mRNA-1273 *n* = 2, BNT162b2 *n* = 5), and in all 17 primary responders, as well as in all 48 HCs (Figure 3A). For those who received a third vaccine in both groups, SARS-CoV-2 antibody titers increased significantly compared to antibody levels measured after the second vaccine dose (median, 2500 (IQR 737–2500)) in LuTRs and ≥2500 in HCs (Figure 3B).

### 7.3. Factors Associated with Seroconversion Rates and Antibody Titers

Potential associations of variables such as age, sex, years since lung transplantation, diagnosis, and immunosuppressive therapy, with seroconversion and antibody titers after COVID-19 mRNA vaccination were considered. In univariate logistic regression, therapy with MMF/MPA was associated with significantly reduced odds for seroconversion after the second vaccine dose (OR 0.11, CI 0.01–0.63, *p* = 0.041). The variables age (in ten-year intervals), sex, and diagnosis of COPD were also associated with reduced odds for seroconversion, but failed to reach statistical significance. (Figure 4A, Table 2, Appendix A) After adjusting the analysis for age and sex, integrating these two variables into multivariate logistic regression models with the other variables, therapy with MMF/MPA was still associated with lower odds for seroconversion, but failed to reach statistical significance. While the diagnosis of cystic fibrosis (CF) was associated with higher odds for seroconversion in the univariate model, adjusting for age and sex, showed the opposite. In both analyses, however, statistical significance was not reached (Figure 4B, Table 2).

### 7.4. Cellular Response to SARS-CoV-2 after the Second Dose of mRNA Vaccine

To assess if SARS-CoV-2 vaccination generated T cell responses, PBMCs collected 1–2 weeks after the second vaccine dose were analyzed using IFN-γ ELISpot assays. T cell responses were induced in 50% (8/16) of LuTRs and 100% (16/16) of HCs. No significant differences in median age in LuTRs with T cells responses (51, 23–64) and without T cell responses (58, 39–71), *p* = 0.24 were detected. Of the eight LuTRs with detectable T cell responses, three (38%) developed humoral responses, whereas five (62%) did not seroconvert after the second vaccination (Figure 5).

### 7.5. Adverse Events

Data of adverse events were systematically documented in all participants. No aggravation of pre-existing symptoms, no change in immunosuppressive medical history, and no serious adverse events were stated during the follow-up. We report no breakthrough SARS-CoV-2 infections in LuTRs within 5 months after two vaccine doses.

## 8. Discussion

This study demonstrates that COVID-19 mRNA vaccine response in LuTRs are significantly reduced compared to healthy controls. We showed a 56% seroconversion rate after two doses of mRNA vaccine in LuTRs, and a clear tendency towards better humoral responses after the third dose. Although the seroconversion rate in our study was considerably reduced, previous studies reported even lower responses after two COVID-19 mRNA vaccine doses, ranging from a complete lack of humoral responsiveness, to a response rate of 39% [9,10].

Different time points of sample collection could probably explain such differences after the second vaccination, as well as demographic factors, such as age, time since transplantation, and differences in immunosuppressive regimens employed. Regarding immunosuppressive therapy, diminished immune responses were mostly reported in patients treated with mycophenolate mofetil or rituximab [8,19,20]. Mycophenolate mofetil, one of the key immunomodulators used by SOTRs, was associated with weak antibody responses to COVID-19 vaccines [9,10,21]. In line with this, reduced antibody responses in SOTRs treated with antimetabolite immunosuppressive therapy have been reported after conventional vaccination [19]. Using multivariate logistic regression analysis, we reveal a lower seroconversion rate in LuTRs treated with MMF/MPA. These findings support some of the findings by Morishita et al., who demonstrated improved seroconversion in SOTRs after withdrawal from MMF/MPA [22].

Cellular immunity after COVID-19 vaccination probably plays a contributing role in controlling SARS-CoV-2 infection, even in the absence of a humoral response [8]. In our study, cellular immunity was analyzed in 16 patients after the second vaccine dose. T cell responses to COVID-19 mRNA vaccination were only detected in 50% of patients; however, in three LuTRs, these were detected even in the absence of humoral responses. Our results for cellular responses are comparable to those published by Hall et al., who found a significant number of patients with a positive SARS-CoV-2 specific CD^4+^ T cell response, often in the absence of a detectable antibody response [14].

Furthermore, we report a decrease in SARS-CoV-2 S antibody levels over time in both study groups. As expected, antibody titers declined after 6 months in most vaccine recipients. Surprisingly, an increase in antibody titers was detected in 14 LuTRs and four HCs, suggesting a delayed immune response in some patients receiving immunosuppressive therapies.

A decline in SARS-CoV-2 antibodies over 5 months was reported after BNT162b2 vaccination in extensive retrospective studies from the US and Israel [23]. Therefore, a third vaccine dose is recommended to improve protection against serious illness from COVID-19, and to reduce the potential of transmission.

In SOTRs requiring continuous immunosuppression, the third dose of the COVID-19 mRNA vaccine has been reported safe and effective. In a randomized trial, a third dose of mRNA vaccine in 120 SOTRs had substantially higher immunogenicity than a placebo [14]. In contrast, in a small study population, Havlin et al. showed impaired humoral responses to the third dose (13%), despite detectable specific T cells responses [24]. In the study published by Hoffman et al., a humoral response after the third dose was detected in 62% LuTRs [25]. In the present study, a humoral response was detected in 22 of 24 (92%) LuTRs, with significantly higher antibody titers compared to those after the second dose.

The limitation of our study is the small sample size, especially for patients in T cell analyses; thus, we could not investigate, in detail, the influence of immunosuppressive therapies on cellular responses. Nevertheless, our data confirm that some LuTRs are able to mount cellular responses even in the absence of seroconversion. Further studies with a higher number of LuTRs are needed to confirm our findings.

COVID-19 vaccines appear to have high efficacy against severe disease and death. Our study detected no breakthrough vaccine infection in LuTRs at a 6-month follow-up. This could be explained by T cell responses in the absence of humoral responses, and by more conscious use of other protective measures.

In summary, our data indicate that LuTRs receiving two doses of the COVID-19 mRNA vaccine have diminished humoral and cellular immune responses. However, after the third dose, both a significantly higher rate of seroconversion, and a considerable rise in SARS-CoV-2 antibody titers, was achieved.

## Figures and Tables

**Figure 1 vaccines-10-01130-f001:**
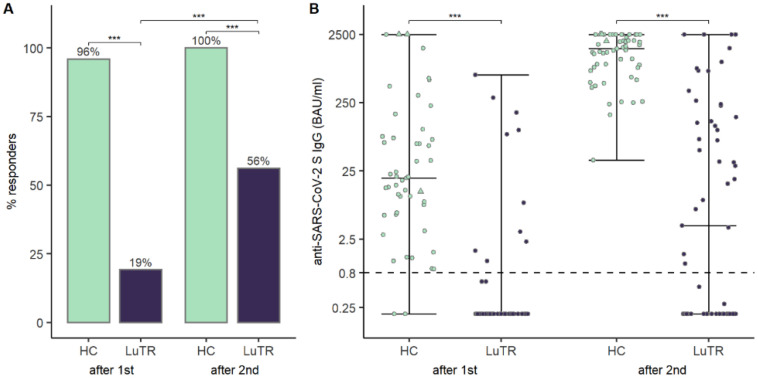
Humoral response after mRNA vaccine in LuTRs and in HCs after the first and second immunization: seroconversion rates of SARS-CoV-2 antibodies and increase in seroconversion rate (% of response) (**A**). SARS-CoV-2 S antibody levels (BAU/mL). Upper level of quantification: 2500 BAU/mL. The horizontal line indicates the cutoff for seroconversion. Circles represent individual antibody levels. Participants with a history of COVID-19, prior to immunization, are represented as triangles (**B**). (*** significant).

**Figure 2 vaccines-10-01130-f002:**
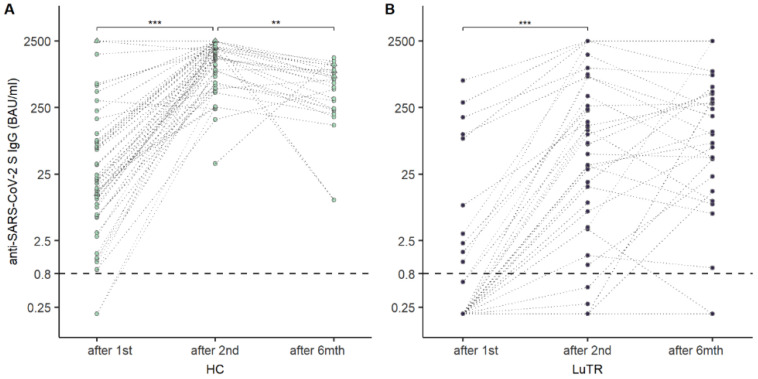
Change in SARS-CoV-2 S antibody levels (BAU/mL) over time in HCs (**A**) and LuTRs (**B**). Upper level of quantification: 2500 BAU/mL. The horizontal line indicates the cutoff for seroconversion. Circles represent individual antibody levels. Participants with a history of COVID-19, prior to immunization, are presented as triangles. Darker lines indicate overlapping points. (**, *** significant).

**Figure 3 vaccines-10-01130-f003:**
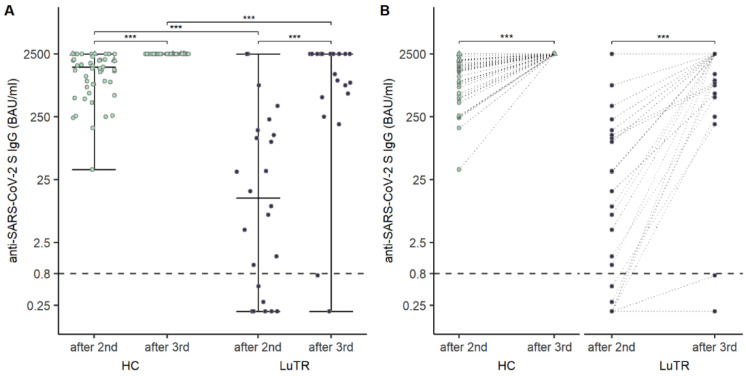
Humoral response in LuTRs and in HCs 4 weeks after the second and the third dose of vaccination: SARS-CoV-2 S antibody levels (BAU/mL) (**A**), and change in SARS-CoV-2 S antibody levels (BAU/mL) after second and third immunization (**B**). The horizontal lines indicate the cutoff for seroconversion. Circles represent individual antibody titers. Participants with a history of COVID-19, prior to immunization, are presented as triangles. Darker lines indicate overlapping points. (*** significant).

**Figure 4 vaccines-10-01130-f004:**
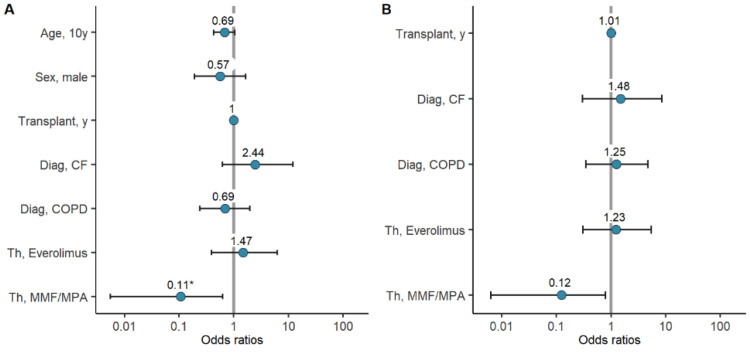
Logistic regression assessing seroconversion in LuTRs: univariate logistic regression including age, sex, years since lung transplantation, diagnosis, immunosuppressive therapy (**A**). Multivariate logistic regression for years since lung transplantation, diagnosis, and immunosuppressive therapy, adjusted for age and sex (**B**). (* significant).

**Figure 5 vaccines-10-01130-f005:**
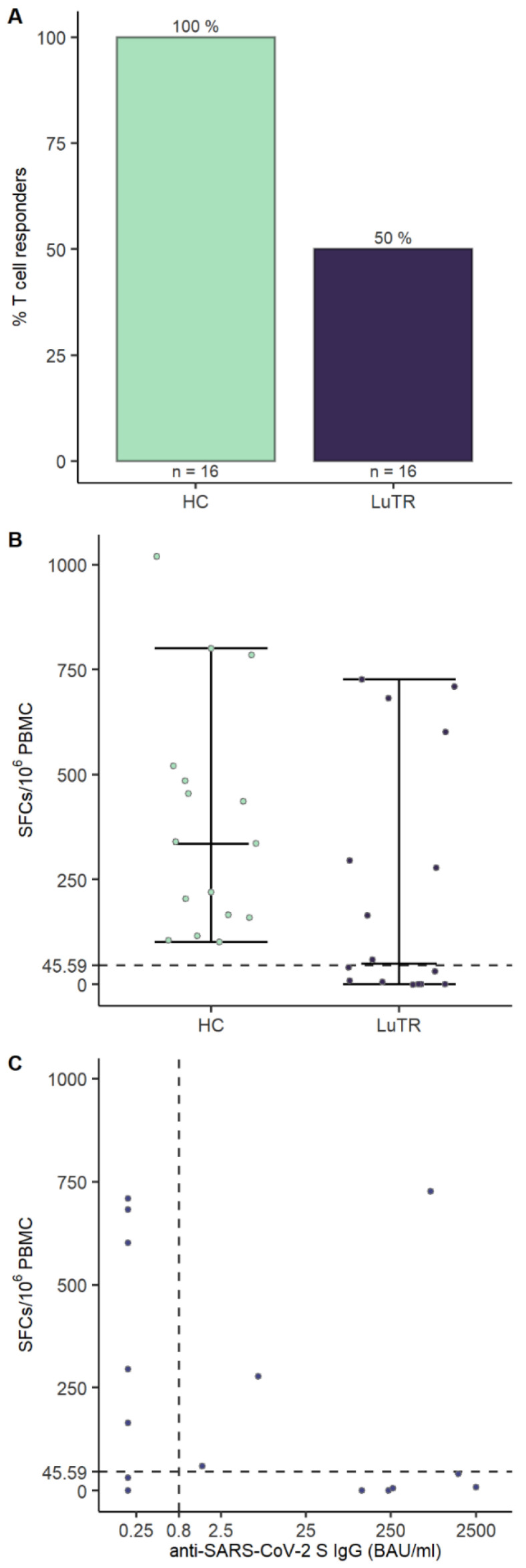
T cell responses to SARS-CoV-2 mRNA vaccination. T cell response rates and magnitudes in LuTRs and HCs: Bars indicate proportion of patients with a T cell response against SARS-CoV-2 peptide pools at 2–4 weeks after second vaccination dose (**A**). Circles represent individual T cell responses; y axis indicates the number of spot-forming cells (SFCs) per 10^6^ PBMCs. Dashed lines indicates mean SFCs per 10^6^ PBMCs + 3 times the standard deviation for spike peptide pool reactivity calculated from pre-pandemic controls (**B**). Scatterplot of humoral and cellular immune responses in LuTRs. The x axis represents Elecsys^®^ Anti-SARS-CoV-2 IgG titers (BAU/mL), with the vertical dashed line indicating the cutoff for seroconversion. The y axis represents SCFs per 10^6^ PBMCs, with the horizontal dashed line indicating mean SFCs per 10^6^ PBMCs + 3 times the standard deviation for spike peptide pool reactivity calculated from pre-pandemic controls (**C**).

**Table 1 vaccines-10-01130-t001:** Demographics.

	LuTRs (*n* = 57)	(*n* = 57)
Age, median (IQR)	55.5 (46.5–58.25)	55 (44.0–64.00)
Sex, female *n* (%)	35 (61)	34 (60)
Time in years between lung organ transplantation and vaccination, median (IQR)	7 (4–11)	NA
Immunosuppressive therapy including prednisone (*n*, %)	57 (100)	NA
Immunosuppressive therapy including tacrolimus (*n*, %)	56 (100)	NA
Immunosuppressive therapy including MMF/MPA *n* (%)	47 (83)	NA
Immunosuppressive therapy including everolimus *n* (%)	11 (19)	NA
Immunosuppressive therapy including ciclosporine *n* (%)	1 (2)	NA
Therapy combination (*n*, %):		
Tacrolimus, MMF/MPA, prednisone	39 (68)	NA
Everolimus, tacrolimus, MMF/MPA, prednisone	7 (12)	NA
Tacrolimus, prednisone	6 (11)	NA
Everolimus, tacrolimus, prednisone	3 (5)	NA
Everolimus, tacrolimus	1 (2)	NA
Ciclosporine, MMF/MPA/prednisone	1 (2)	NA
Therapy number *n* (%):		NA
2 Agents	7 (12)	NA
3 Agents	43 (75)	NA
4 Agents	7 (12)	NA
Drug levels (median, IQR):		
Ciclosporine (*n* = 1)	120	NA
Everolimus (*n* = 8)	3.4 (2.7–3.6)	NA
Tacrolimus (*n* = 52)	5.6 (4.3–6.8)	NA
Dose prednisone (median, IQR, *n* = 56)	5 (5–5)	NA
Vaccine *n* (%) 2 doses *n* (%)	57 (100)	57 (100)
mRNA-1273	2 (4)	0 (0)
BNT162b2	55 (96)	57 (100)
Vaccine *n* (%) 3 doses *n* (%)	24 (42)	53 (93)
mRNA-1273	2 (8)	2 (4)
BNT162b2	19 (80)	51 (96)
ChAdOx1 nCoV-19	2 (8)	0 (0)
Ad26.COV2.S	1 (4)	0 (0)

Mycophenolate mofetil/mycophenolic acid (MMF/MPA); Not Applicable (NA).

**Table 2 vaccines-10-01130-t002:** Univariate and multivariate logistic regression.

Univariate Logistic Regression			Multivariate Logistic Regression ^§^	
	OR (CI 95%)	*p*-Value	OR (CI 95%)	*p*-Value
Age, ten years	0.69 (0.43–1.04)	0.088	—	—
Sex, male	0.6 (0.2–1.74)	0.345	—	—
Years since transplant	1 (0.92–1.1)	0.985	1.01 (0.92–1.12)	0.793
Diagnosis, CF	2.05 (0.5–10.43)	0.337	0.81 (0.13–5.38)	0.820
Diagnosis, COPD	0.69 (0.24–1.98)	0.495	1.49 (0.4–6.03)	0.563
Therapy, everolimus	1.47 (0.39–6.26)	0.578	1.28 (0.32–5.69)	0.729
Therapy, MMF/MPA	0.11 (0.01–0.63)	0.041	0.12 (0.01–0.78)	0.059

^§^ Logistic regression models for diagnosis of CF (cystic fibrosis), diagnosis of COPD (chronic obstructive pulmonary disease), therapy with everolimus, and therapy with MMF/MPA (mycophenolate mofetil/mycophenolic acid), each adjusted for age and sex.

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
