# Peer review of "Immune Response after mRNA COVID-19 Vaccination in Lung Transplant Recipients: A 6-Month Follow-Up"

_vaccines, 2022, doi:10.3390/vaccines10071130_

Round 1
Reviewer 1 Report
This article is very interesting and explains how immune responses can be enhanced against SARS-Cov2 in immune suppressed patients through different vaccination regimens. I have few concerns about this article. Below are the details,
1. Table 1, it would be important if the author shows the distribution of patients in a graph to show the number of patients closure to lower margin of age and number of patients towards the higher margin of age in both the groups.
2. Table 1, the immune responses for mRNA-1273 can be different from the immune responses for BNT162b2 vaccine. The author could include the immune responses for mRNA-1273 separately.
3. Figure 4, it would be necessary to know the ages of the patients used for IFN-y ELISpot to check for T cell immune responses since age could interfere with T cell characteristics.
4. Figure 5, it would be important to know the age distribution of the patients used in the antibody analysis post 2nd and 3rd dose of vaccine since it would help readers to understand how age also affects the antibody responses.
Author Response
Dear Editor,
Thank You very much for all suggestions for improving our manuscript. We have tried to answer them,
Enclosed are all the answers (point to point)
Kind regards,
Selma Tobudic
This article is very interesting and explains how immune responses can be enhanced against SARS-Cov2 in immune suppressed patients through different vaccination regimens. I have few concerns about this article. Below are the details,
- Table 1, it would be important if the author shows the distribution of patients in a graph to show the number of patients closure to lower margin of age and number of patients towards the higher margin of age in both the groups.
Response:
We added Suppl. Figure 1. with age distribution of LUTRs and healthy controls.
- Table 1, the immune responses for mRNA-1273 can be different from the immune responses for BNT162b2 vaccine. The author could include the immune responses for mRNA-1273 separately.
Response:
We included only two patients, who received the mRNA-1273 vaccine, and we could not make a comparative analysis of the immune response in patients after the BNT162b2 vaccine. However, we added in the text the immune response of two patients who received the mRNA-1273 vaccine
Line 184-186: After the third dose of the Covid-19 vaccine, seroconversion was achieved in the LuTRs group in 5 of 7 primary non responders (mRNA-1273 n=2, BNT162b2 n=5) and in 17 of 17 primary responders, as well as in 48 of 48 HCs.
3.Figure 4, it would be necessary to know the ages of the patients used for IFN-y ELISpot to check for T cell immune responses since age could interfere with T cell characteristics.
Response:
Age probably plays a crucial role in the humoral and cellular response. We could not see any significant difference in median age between LuTRs with and without T-cell responses in this study, however more extensive studies are probably needed to test this hypothesis.
Line: 210-212 No significant difference in median age in LuTRs with T-cells responses (51, 23-64) and without T-cell responses (58, 39-71), p=0.24 was detected.
- Figure 5, it would be important to know the age distribution of the patients used in the antibody analysis post 2nd and 3rd dose of vaccine since it would help readers to understand how age also affects the antibody responses.
Response:
189-194 potential association of variables such as age, sex, years since lung transplantation, diagnosis and immunosuppressive therapy with seroconversion and antibody titers after Covid-19 mRNA vaccination was considered. In univariate logistic regression, therapy with MMF/MPA was associated with significantly reduced odds for seroconversion after the second vaccine dose (OR 0.11, CI 0.01 – 0.63, p = 0.041). The variables age (in ten year intervals), sex and the diagnosis COPD were also associated with reduced odds for seroconversion, but failed to reach statistical significance. (Figure 3A, Table 2, and Suppl Figure1.)
Reviewer 2 Report
Authors performed a prospective study on the humoral responses of COVID-19 vaccination in lung transplant recipients. The study is interesting and the results were rational. I have several concerns regarding the article:
1. The size of cohort (n=57) is relatively small. I am concerning what this could bring for a meaningful conclusion based on the size and information gleaned.
2. Immune response could be from a variety of reasons. Between two cohorts, the authors can not assess the exact effect of immunosuppressing drugs on patients' immune responses, especially the acquired immunity? If so, the difference between the two cohorts needs to be carefully analyzed.
3. Can the authors provide the infection profiles of all the patients? That is, whether those participating patients have been infected by SARS-CoV-2 or its variants?
4. References could be updated with more new related studies.
Author Response
Dear Editor,
Thank You very much for all suggestions for improving our manuscript. We have tried to answer them,
Enclosed are all the answers (point to point)
Kind regards,
Selma Tobudic
Authors performed a prospective study on the humoral responses of COVID-19 vaccination in lung transplant recipients. The study is interesting and the results were rational. I have several concerns regarding the article:
- The size of cohort (n=57) is relatively small. I am concerning what this could bring for a meaningful conclusion based on the size and information gleaned.
- Immune response could be from a variety of reasons. Between two cohorts, the authors can not assess the exact effect of immunosuppressing drugs on patients' immune responses, especially the acquired immunity? If so, the difference between the two cohorts needs to be carefully analyzed.
Response:
We added a small size of the study population as a limitation of this study. However, our results correlate with other studies investigating humoral response in solid organ recipients.
- Can the authors provide the infection profiles of all the patients? That is, whether those participating patients have been infected by SARS-CoV-2 or its variants?
Response:
Line 219-220 We report no breakthrough SARS-CoV-2 infections in LuTRs within 5 months after two vaccine doses.
- References could be updated with more new related studies.
Response:
Due dynamic process in Covid-19 research, it is not easy to insert all relevant References. We try to update references with a few new published.